# Wind Field Retrieval with Rain Correction from Dual-Polarized Sentinel-1 SAR Imagery Collected during Tropical Cyclones

**Weizeng Shao [1,2,\*], Zhengzhong Lai [1], Ferdinando Nunziata [3], Andrea Buono [3], Xingwei Jiang [4,5] and Juncheng Zuo [1,2]**

1 College of Marine Sciences, Shanghai Ocean University, Shanghai 201306, China
2 Key Laboratory of Marine Ecological Monitoring and Restoration Technologies, Ministry of Natural Resources, Shanghai 201206, China
3 Dipartimento di Ingegneria, Università degli Studi di Napoli Parthenope, 80143 Napoli, Italy
4 Southern Marine Science and Engineering Guangdong Laboratory (Guangzhou), Guangzhou 511458, China
5 National Satellite Ocean Application Service, Ministry of Natural Resources of the People's Republic of China, Beijing 100081, China
\* Correspondence: wzshao@shou.edu.cn; Tel.: +86-21-1900326

**Abstract:** The purpose of this study is to include rain effects in wind field retrieval from C-band synthetic aperture radar (SAR) imagery collected under tropical cyclone conditions. An effective and operationally attractive approach to detect rain cells in SAR imagery is proposed and verified using four Sentinel-1 (S-1) SAR images collected in dual-polarized (vertical-vertical (VV) and vertical-horizontal (VH)) interferometric-wide swath imaging mode during the Satellite Hurricane Observation Campaign. SAR images were collocated with ancillary observations that include sea surface wind and rain rate from the Stepped-Frequency Microwave Radiometer (SFMR) on board of the National Oceanic and Atmospheric Administration aircraft. The winds are inverted from VV- and VH-polarized S-1 image using the CMOD5.N and S1IW.NR geophysical model functions (GMFs), respectively. Location and radius of cyclone's eye, together with the TC central pressure, are calculated from the VV-polarized SAR-derived wind and a parametric model. A cost function is proposed that consists of the difference between the measured VV-polarized SAR normalized radar cross section (NRCS) and the NRCS predicted using CMOD5.N forced with the wind speed retrieved by the VH-polarized SAR images using S1IW.NR GMF and the wind direction retrieved from the patterns visible in the SAR image. This cost function is related to the SFMR rain rate. Experimental results show that the difference between measured and predicted NRCS values range from 0.5 dB to 5 dB within a distance of 100 km from the cyclone's eye, while the difference increases spanning from 3 dB to 6 dB for distances larger than 100 km. Following this rationale, first the rain bands are extracted from SAR imagery and, then, the composite wind fields are reconstructed by replacing: (1) dual-polarized SAR-derived winds over the rain-free regions; (2) winds simulated using the radial-vortex model over the rain-affected regions. The validation of the composite wind speed against SFMR winds yields a < 2 m s$^{-1}$ and > 0.7 correlation (COR) at all flow directions up to retrieval speeds of 70 m s$^{-1}$. This result outperforms the winds estimated using the VH-polarized S1IW.NR GMF, which call for high error accuracy, such as about 4 m s$^{-1}$ with a 0.45 COR ranged from 330° to 360°.

**Keywords:** wind retrieval; rain correction; tropical cyclones

## 1. Introduction

Tropical cyclones (TCs) always result in strong winds, extreme waves and heavy rainfall, which are significant disasters on near-shore waters [1]. The met-ocean conditions associated to TCs make real-time observations from ships and moored buoys unfeasible.

Instead, remote-sensed technology is useful for rainfall monitoring [2]. The Tropical Rainfall Measuring Mission (TRMM)—a well-known international cooperation project designed for global precipitation observations—operating at microwaves has released three-dimensional data since November 1997 [3,4]. However, remote-sensed TRMM rainfall data are not suitable for the meso- and small-scale studies concerning the air–sea interactions due to the relative coarse spatial resolution of approximately 0.25° grid (about 25 km).

At present, synthetic aperture radar (SAR) images is an active and fine spatial resolution (down to 1 m) sensor with an all-day and almost all-weather capability of sea surface monitoring in polar region [5] and TCs, e.g., the morphology of cyclones [6], strong winds [7,8] and extreme waves [9]. Rain footprints over sea surface were also qualitative observed in SAR imagery [10]. Traditionally, in SAR images, three main aspects rule the rain-induced effects on sea surface roughness [11,12]: disturbances on wind-generated short waves winds [13,14]; splash products, i.e., volume scattering and attenuation due to raindrops at the air-sea layer [15]; large-scale downdraft roughness modulated by rain-induced wind flow [16]. Accordingly, rain cells affect the homogeneity of the SAR image distribution and the normalized radar cross-sections (NRCS) measured by the SAR. Even though the rain-induced characteristics on co-polarized SAR backscattering at C- and X-band was investigated [17,18], the rain attenuation of the SAR wind signals at co- and cross-polarized channel is complicated. Moreover, the correction of rain cells is needed for SAR-derived wind retrieval under extreme weather using a practical scheme.

The wind retrieval methodology from SAR image is well developed in last decades [8,19]. SAR-based wind retrieval relies on the relationship between sea surface wind and co- and cross-polarized SAR measurements [20], through the development of geophysical model functions (GMFs) [21–23]. The validation of wind retrieval against the measurements from scatterometer [24] and moored buoys yields a 2 m s$^{-1}$ accuracy of wind speed using co-polarized C-band [25,26] and X-band SAR [27] GMF. However, it is revealed that co-polarized GMFs suffer the saturation at high winds, i.e., wind speed larger than 25 m s$^{-1}$ [28]. Utilizing the RADARSAT-2 (R-2) fine quad mode data collocated with operational weather buoys [29], it is found that the cross-polarized SAR backscattering signal is linearly related with wind speed up to 55 m s$^{-1}$ [30,31] and is independent of incidence angle and wind direction. Although cross-polarized GMFs work well at very high winds [7,32], the accuracy of wind field retrieval from cross-polarized NRCS SAR measurements at low-to-moderate winds is strictly related to the signal-to-noise ratio [33], which is particularly important for low-backscatter cross-pol measurements at small incidence angles.

In this study, four dual-polarized Sentinel-1 (S-1) C-band SAR images are considered, which were taken by the Satellite Hurricane Observation Campaign (SHOC), providing high-quality data for wind [18] and wave research [34] in TC. These images were collocated with the wind field and rain rate observations provided by the Stepped-Frequency Microwave Radiometer (SFMR) on-board of the National Oceanic and Atmospheric Administration (NOAA) aircraft. The TC parameters, including location of cyclone's eye, central pressure, maximum wind speed and maximum wind radius, are calculated based on the VV-polarized SAR wind retrieval algorithm and a parametric model developed by Holland (1980). A practical approach for the identification of rain cells is developed when analyzing the cost function of the SAR backscattering related to the SFMR rain rate. The information on TCs is used to produce a wind field using a radial-vortex model [35] and the regions identified as raining is replaced by the simulated wind field. Finally, a composite wind field is derived from S-1 SAR-derived wind field at VV- and VH- polarization channel after taking the corrections from the radial-vortex modeling wind field.

The remainder of this paper is organized as follows: the dataset is briefly described in Section 2, including the four dual-polarized S-1 IW SAR images collocate with the SFMR measurements in TC and the SAR-derived wind fields from VV-polarized and VH-polarized images. Besides, the methodology of the rain cell identification from SAR data, estimation of TC parameters and the parametric wind profile model is introduced in Section

2. Section 3 presents the retrieved wind speeds after rain corrections. The conclusions are summarized in Section 4.

## 2. Materials and Methods

### 2.1. Materials

In this study, four S-1 SAR images acquired in dual-polarized (VV+VH) interferometric-wide (IW) swath imaging mode during TCs are considered with the clear cyclone eye, see Table 1. They are collocated with sea surface wind and rain rate observations obtained by SFMR, which benefits the comprehensive TC studies using SAR data [36]. The SFMR provide near real-time observations of the wind speed and rain rate within and around tropical cyclone [37]. The SAR wind retrieval is essential for the identification of rain cell. The well-developed GMF at C-band, denoted as CMOD4 [38], has been used for SAR wind retrieval since 1997. The performance of GMF is improved by the updated CMOD family [39] using the following formulation:

$$\sigma^0 = B_0(U_{10}, \theta)(1 + B_1(U_{10}, \theta)\cos\phi + B_2(U_{10}, \theta)\cos 2\phi) \quad (1)$$

where $\sigma^0$ is the linear NRCS measured by SAR, $U_{10}$ is the wind speed at 10-m height, $d$ $\varphi$ represents the angle between the wind direction and the radar look direction and matrix $B$ ($B_0$, $B_1$, and $B_2$) are functions of the incidence angle $\theta$ and the wind speed at 10 m above the sea surface. Because SAR operates using a single look radar beam, a wind vector is impossible to be simultaneously retrieved due to two unknown variables (wind speed and direction) in the GMF. In practice, the wind direction is extracted from the two-dimensional SAR intensity spectrum at wavelengths between 800 m and 3000 m [40] or directly measured following the wind streaks [41]. In this study, the wind directions are estimated from wind-induced low-frequency SAR image features and the 180° ambiguity is removed using external European Centre for Medium-Range Weather Forecasts (ECMWF)-interim (ERA-5) data with a spatial resolution of approximately 0.25° grid. The complete field of wind directions that will be considered in the SAR wind speed retrieval process is obtained using the interpolation method proposed in [42] and we do not repeat it here.

**Table 1.** Main SAR and ancillary information of the dataset.

| TC | Category | Date and UTC Time of the SAR Acquisitions | Pixel Size, Range × Azimuth (m) | Maximum Central Pressure (hPa) | Maximum Wind Speed Radius (km) |
|---|---|---|---|---|---|
| Irma | 4 | 07-09-2017 10:30 UTC | 10 × 10 | 921 | 35 |
| Dorian | 5 | 30-08-2019 22:46 UTC | 10 × 10 | 949 | 54 |
| Lsaias | 1 | 02-08-2020 23:19 UTC | 10 × 10 | 995 | 111 |
| Delta | 4 | 08-10-2020 00:07 UTC | 10 × 10 | 973 | 73 |

As a showcase, the quick-looks of the VV-polarized S-1 SAR images belonging to the dataset are shown in Figure 1, where subfigures refer to TCs: (a) Irma at 10:30 UTC on 7 September 2017, (b) Dorian at 22:46 UTC on 30 August 2019, (c) Lsaias at 23:19 UTC on 2 August 2020 and (d) Delta at 00:07 UTC on 8 October 2020. The red lines represent the SFMR tracks. Recently, aimed to wind retrieval from VH-polarized S-1 SAR images after thermal noise removal, an algorithm termed as S-1 IW Mode Wind Speed Retrieval Model

after Noise Removal (S1IW.NR) was developed [43,44] and the specific formulation is expressed as follows, considering the different performance at various incidence angle $\theta$, e.g., IW1: [31.0°–35.9°], IW2: [35.9°–41.3°] and IW3: [41.3°–46.0°]:

$$\sigma_0^{VH}[dB]=\begin{cases}0.22U_{10}-29.68 & \text{IW1}\\4.67U_{10}^{0.39}-41.02 & \text{IW2}\\-56.67U_{10}^{-0.26} & \text{IW3}\end{cases} \quad (2)$$

where $\sigma_0^{VH}$ is the VH-polarized NRCS united in dB and $U_{10}$ is the wind speed at 10-m height. The cross-polarized GMF is conveniently applied for SAR strong wind retrieval without be prior information on wind direction. Although maximum wind speeds that can be accurately retrieved from VH-polarized SAR images could be up to 50 m s⁻¹ [7,8], the rain effect is not accounted for in the inversion algorithm and the retrieval accuracy is reduced compared to that from VV-polarized SAR images at low-to-moderate winds due to the low signal-to-noise ratio. The inverted four wind maps obtained from VV-polarized S-1 SAR images using the CMOD5.N GMF and VH-polarized S-1 SAR images using the S1IW.NR are shown in Figures 2 and 3, respectively. It can be clearly observed that the VH-polarized SAR-derived wind speeds are significantly greater than retrieval results from VV-polarized SAR images. The statistical analysis of Taylor diagram between VH-polarized SAR-derived wind speeds and the available SFMR measurements in term of flow directions of TC eyes is presented in Figure 4. The flow direction discrete at interval of a 30°. It is found that maximum root mean square error (RMSE) of wind speed is about 4 m s⁻¹ with a 0.45 correlation (COR) ranged from 330° to 360°; however, the RMSE of wind speed is less than 3 m s⁻¹ at other flow directions. Although this result suggests that, in this study, the VH-polarized SAR-derived wind speeds could be applied up to 70 m s⁻¹ due to the retrieval floor, the accuracy is still further improved at special flow directions.

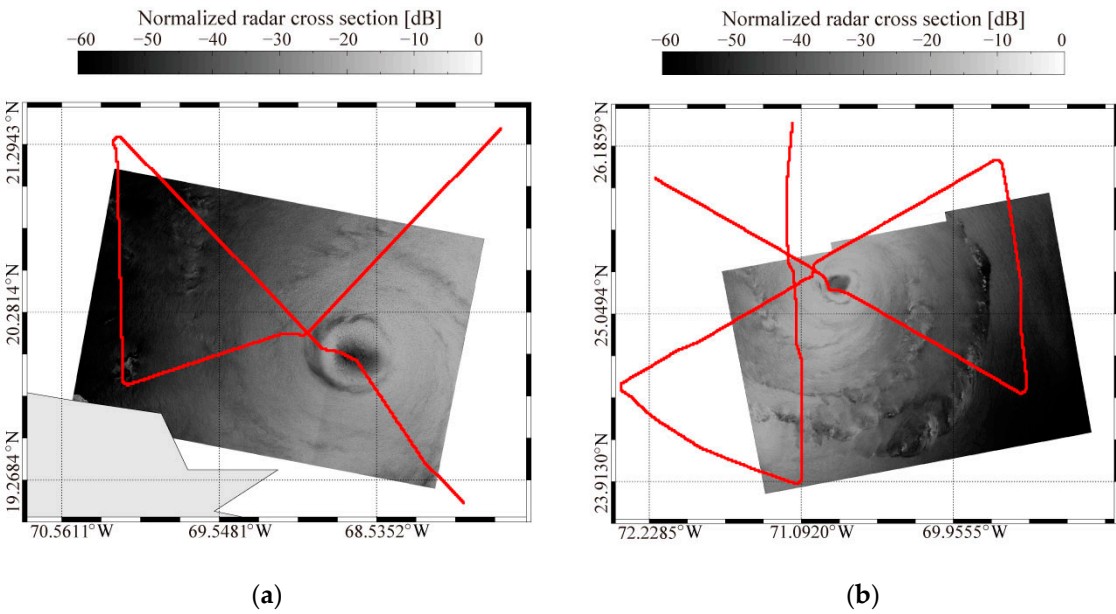

(**a**)            (**b**)

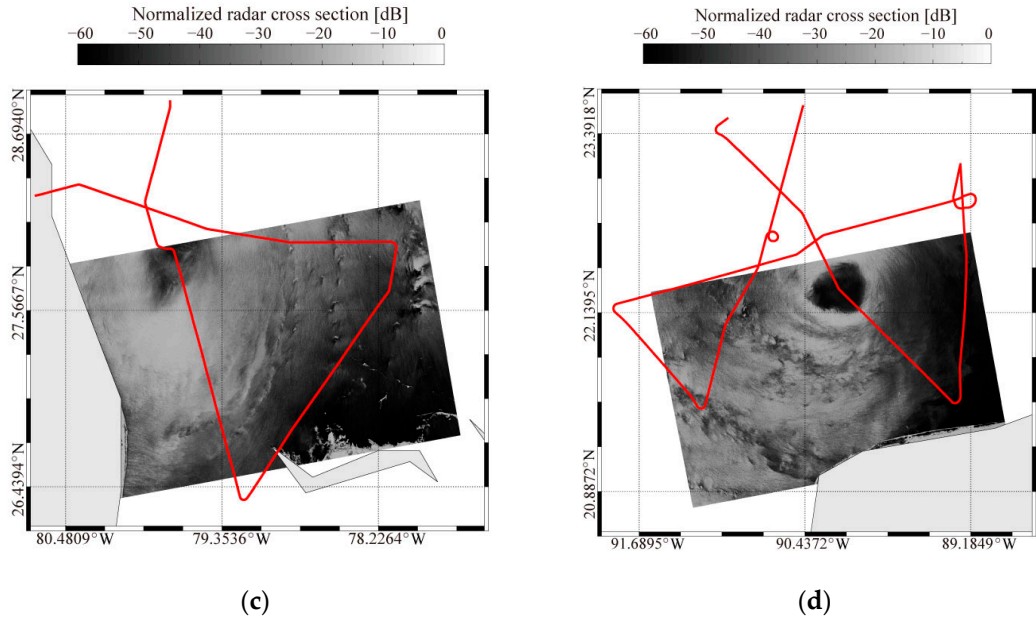

(**c**)                                                                                         (**d**)

**Figure 1.** VV-polarized normalized radar cross section (NRCS) images relevant to the Sentinel-1 (S-1) synthetic aperture radar (SAR) dataset collected over the tropical cyclones (TCs): (**a**) Irma; (**b**) Dorian; (**c**) Lsaias and (**d**) Delta. The red lines represent the he Stepped-Frequency Microwave Radiometer (SFMR) tracks provided by the National Oceanic and Atmospheric Administration aircraft.

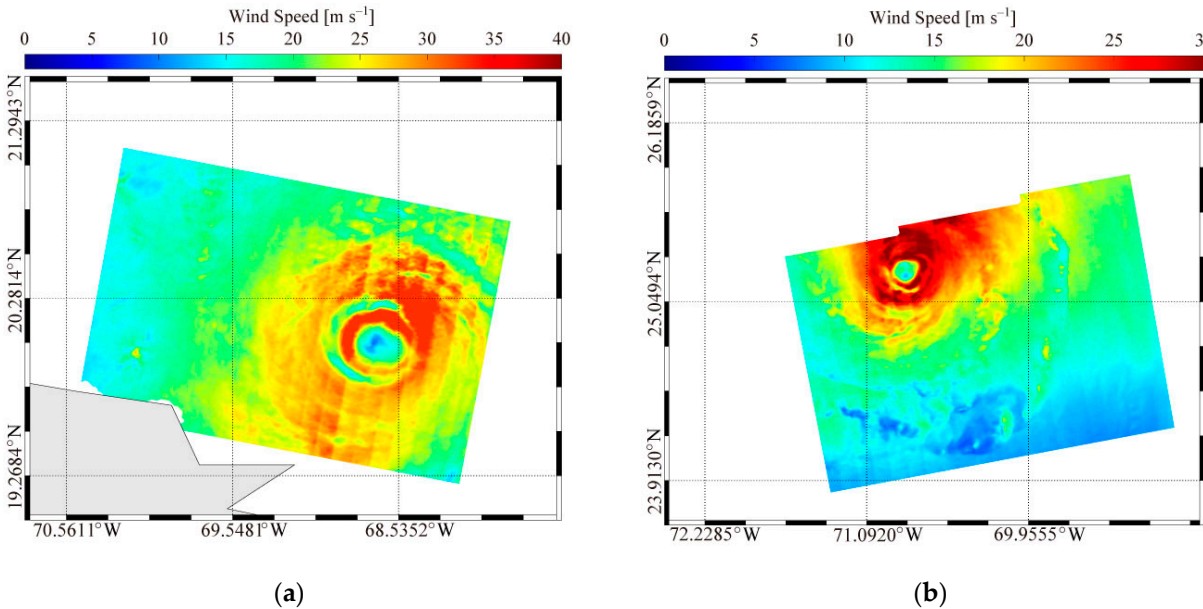

(**a**)                                                                                         (**b**)

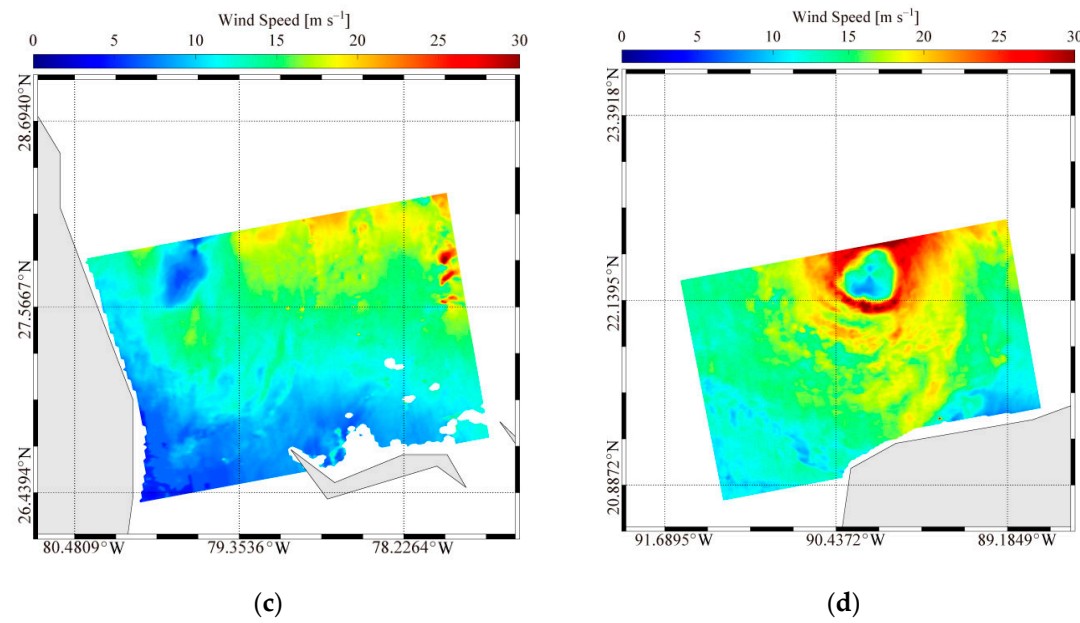

**Figure 2.** Inverted wind maps derived from of VV-polarized S-1 SAR images using the CMOD5.N geophysical model function (GMF). (**a**–**d**) The SAR scenes shown in Figure 1a–d.

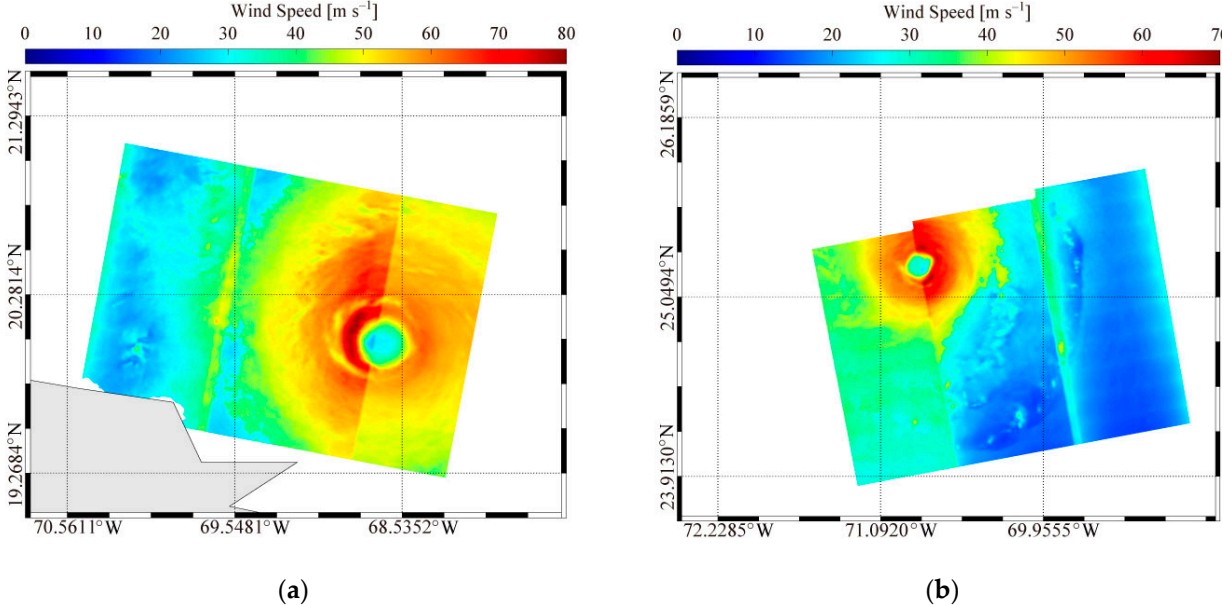

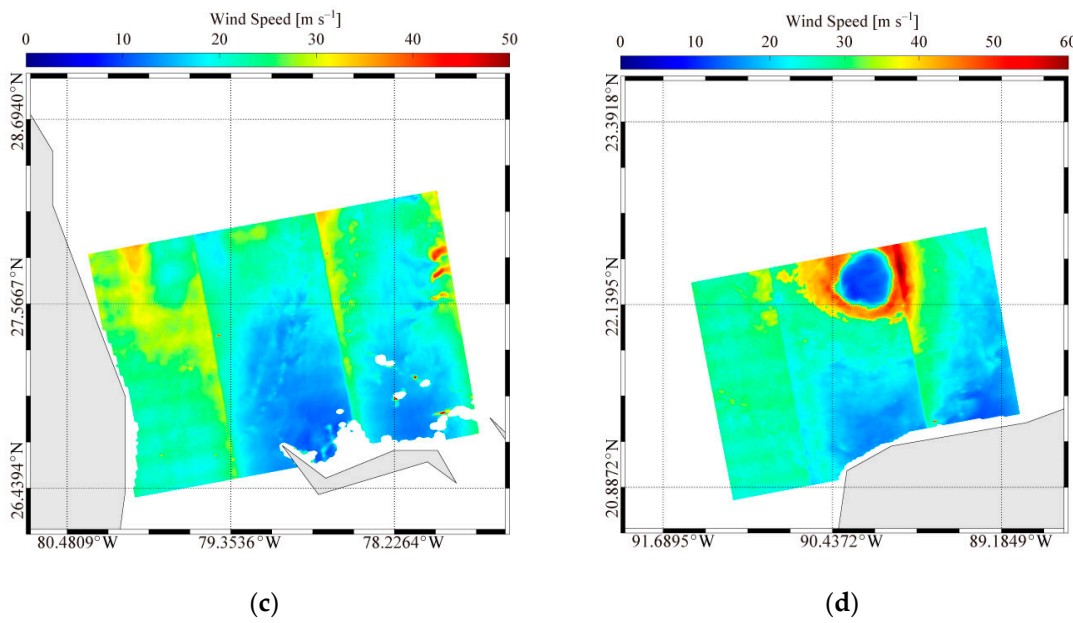

(**c**)　　　　　　　　　　　　　　　　　　　　　　(**d**)

**Figure 3.** Inverted wind maps derived from of VH-polarized S-1 SAR images using the S1IW.NR GMF. (**a**–**d**) The SAR scenes shown in Figure 1a–d.

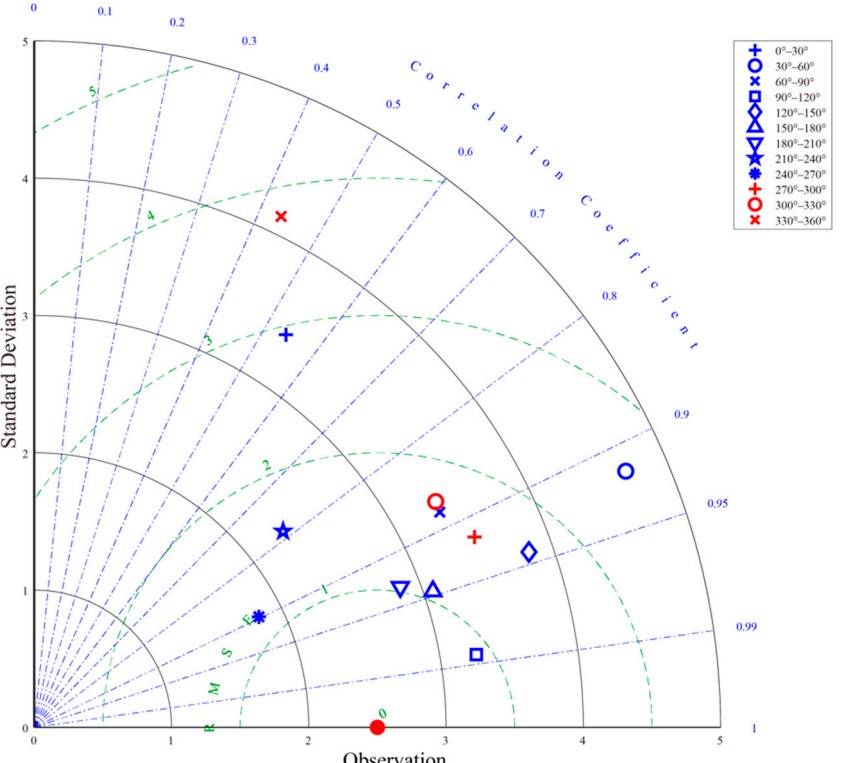

**Figure 4.** Taylor diagram of the matchups between VH-polarized SAR-derived wind speeds and SFMR observation.

### 2.2. Method

In this section, the methodology proposed to identify rain cells and to develop TC wind maps from dual-polarized S-1 SAR imagery is described. Since the SAR-derived wind fields with rainfall parameter are not reliable due to the attenuations in dual-polarized SAR imagery, we used the Holland model to construct the TC wind profile.

### 2.2.1. Rain Cells Identification

The rainfall affects significantly the microwave signal interacting with rain cells. Moreover, the attenuation due to the rainfall is polarization selective, i.e., the co-polarized backscatter exhibits a sensitivity to the rainfall larger than the cross-polarized one. A recent study proposed by [17] showed that the radar backscattering increases with rain rate at low-to-moderate winds. The basic principle is based on the complementary sensitivity of the SAR signal induced by rain cell. Following this rationale, the scheme of detecting rain in SAR data is designed by a complicated cost function relying on four parameters derived from the local gradient of SAR intensity image [45]. Besides, a simple approach is proposed in [35], defined as the difference between the measured VV-polarized NRCS $\sigma_{VV}^0$, and the predicted one $\sigma_S^0$; however, the influence of radar incidence angle is excluded in this method. This latter one is predicted using the CMOD5.N that is forced with a wind speed retrieved from the VH-polarized SAR scene using the S1IW.NR and a wind direction derived from the joint analysis of the pattern visible in the SAR image and using ERA-5 winds. Once the cost function is defined, a threshold value $T$ is set as follows:

$$T = \left| \sigma_{VV}^0 - \sigma_S^0 \right| \ [\text{dB}] \tag{3}$$

To summarize, the following three steps are needed: (1) wind speed retrieval from VH-polarized SAR image using GMF S1IW.NR; (2) prediction of the GMF CMOD5.N-simulated NRCS using SAR-derived wind direction and VH-polarized wind speed retrieved as in (1); (3) applying Equation (3) in term of radar incidence angle from SAR image to calculate the $T$. It is noted that the simple identification method has worse performance than that using local gradient method based on backscattering difference between co-polarized and cross-polarized S-1 image [46], because the information on weak rainfall at regular sea state is missing using simple identification method. However, the efficiency of the approach herein has been confirmed using R-2 images in hurricanes, where rainfall is strong at such a condition.

### 2.2.2. Estimation of TC Parameters

Although the rain flag is a valuable assessment for rain monitoring, quality-flagged wind fields are usually required for TC analyses, i.e., assimilating SAR-derived winds into comprehensive numerical prediction models and the distribution of extreme waves produced by high winds. Within this framework, rainfall corrections on the rain-flagged winds are needed to obtain quick estimations of the complete TC wind field. The well-known parametric Holland wind model [47] is an analytical model for the radial profile of wind speed under TC conditions:

$$V_g = \left[ \frac{AB(p_n - p_c)e^{-A/r^B}}{\rho r^2} + \frac{r^2 f^2}{4} \right]^{1/2} - \frac{rf}{2} + V_{TC} \tag{4}$$

where $V_g$ is the wind speed at radius $r$, $p_c$ and $p_n$ are the central and ambient pressure, respectively, $f$ is the Coriolis force parameter, $\rho$ is the air density constant, $V_{TC}$ is the movement velocity of TC and the parameters $A$ and $B$ represent the location relative to the origin and the shape related to the maximum wind radius $r_{max}$ and maximum wind radius $v_{max}$:

$$A = r_{max}^B \tag{5}$$

$$B = \rho e^{\frac{v_{max}^2}{p_n - p_c}} \tag{6}$$

In this study, the method proposed in [48] to estimate TC parameters is employed. It is a three-step procedure that consists of: (1) an edge detection technique based on the "Daubechies D4" wavelet applied on the VV-polarized NRCS SAR image to determine the TC's eye, because the eye of the TC is detected as a low-frequency feature at this scale;

(2) select two cuts in the range and azimuth directions through TC eyes and the average distance of the two peak values in each cut, that is maximum wind radius $v_{max}$; (3) the wind speed retrieval using the CMOD5.N GMF and the selection of regions where wind speed is lower than 20 m s⁻¹ which are less affected by rain; (4) the minimization of the cost function:

$$J(p_\sigma \ v_{max}) = \sum_{ij} |\sigma^0_{VV,ij} - \sigma^0_{Holland,ij}(p_\sigma \ v_{max})| \tag{7}$$

where $\sigma^0_{Holland,ij}(p_c, v_{max})$ is the simulated NRCS by CMOD5.N GMF using Holland winds and $\sigma^0_{VV,ij}$ is the VV-polarized NRCS measured by the SAR. Results relevant to the comparison between SAR-derived TC intensity, e.g., central pressure $p_c$, maximum wind speed $v_{max}$ and maximum wind radius $r_{max}$, and the SFMR observations are presented in Figure 5. It is found that the RMSEs of central pressure and maximum wind speed are 15.6 hPa and 10.8 m s⁻¹, respectively, comparing the retrieval results with the best-track information of NOAA.

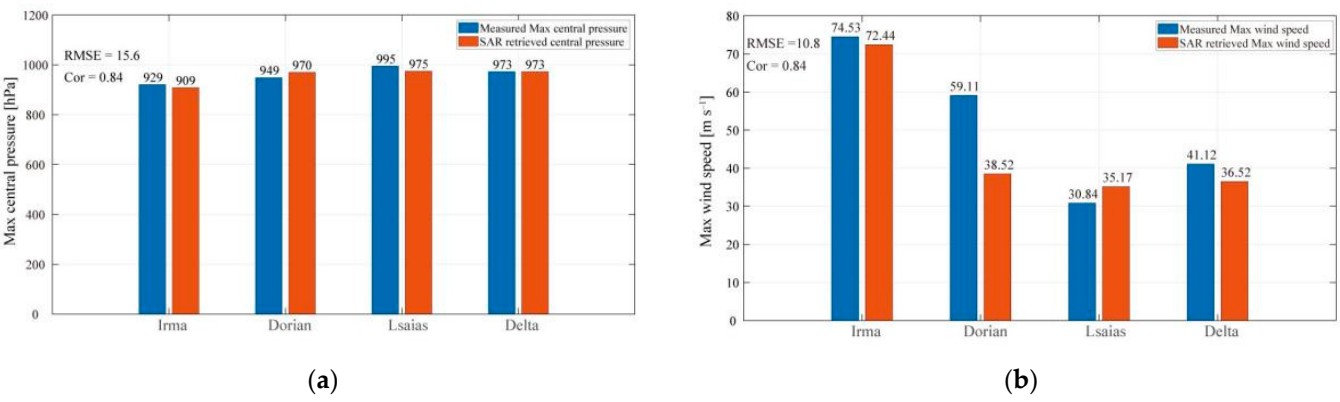

(**a**)    (**b**)

**Figure 5.** Comparison between SAR-derived TC parameters and SFMR observations. (**a**) Central pressure and (**b**) maximum wind speed.

### 2.2.3. TC Wind Radial Profile Model

A TC is mesoscale cyclonic system with a low-pressure center and strong wind speeds around an eyewall, which maintains a unique wind profile along the radial direction. According to the rational physics, the TC wind profile has strict relation with the strength of the TC [47]. This kind of radial profile model has been widely used to reconstruct cyclostrophic wind field as knowing TC best tracks. The revised radial profile model was recently developed [49] consisting of an outer wind and surface pressure, sea surface temperature, radius of maximum winds, and central pressure. Less additional information is required in the SAR wind retrieval process; thus, the following equation is employed for reconstructing radial wind filed:

$$V_r = \begin{cases} v_{max}\left(\dfrac{r}{r_{max}}\right) & r < r_{max} \\ v_{max}\left(\dfrac{r}{r_{max}}\right)^{0.5} & r \geq r_{max} \end{cases} \tag{8}$$

where $V_r$ is the radial wind speed at the distance $r$ from TC eye. As mentioned in [35], the parameters in Equation (8) were calculated by fitting the function to the VH-polarized SAR wind data along each radial direction by a least square method depending on the option of radial direction. In this study, the maximum wind speed $v_{max}$ and maximum wind radius $r_{max}$ are estimated using the method in the above section.

Collectively, the composite wind field after rain correction is reconstructed by VV-polarized SAR winds retrieved using CMOD5.N, VH-polarized SAR winds retrieved using S1IW.NR and the vortex-modeling TC wind field. The flowchart of the methodology is depicted in Figure 6.

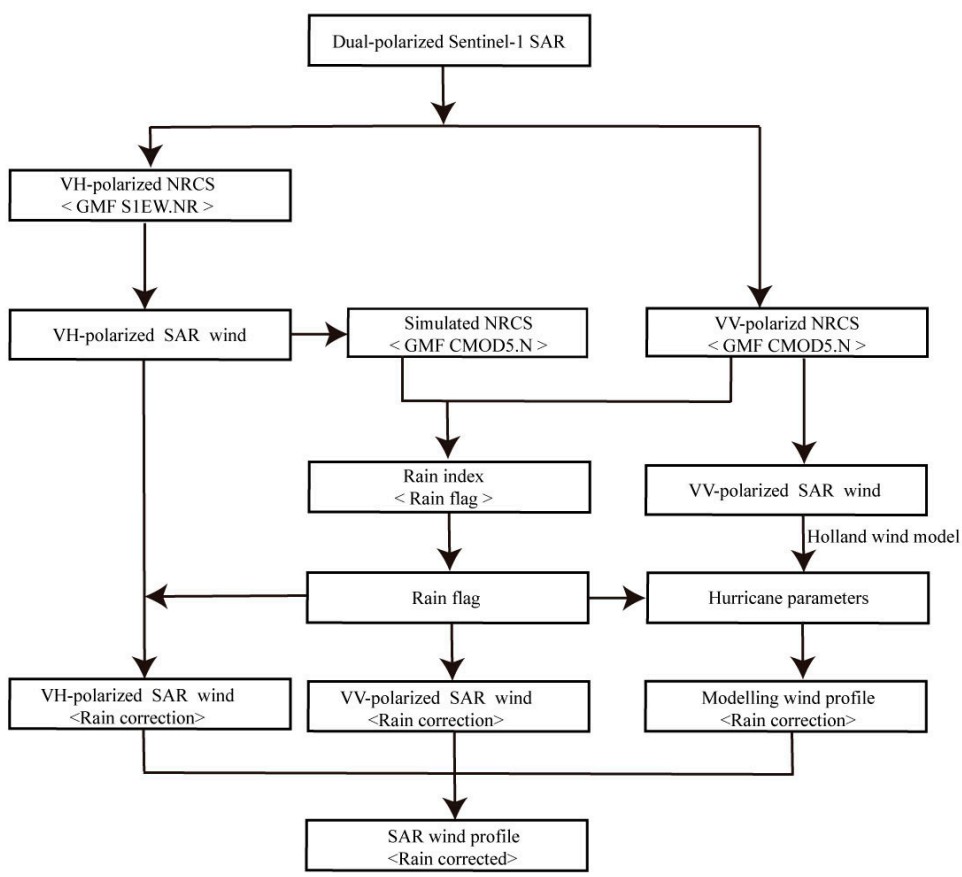

**Figure 6.** Flowchart of the proposed methodology.

## 3. Results

In this section, the identification results of rain cell for four S-1 SAR image in TCs are first presented. Then, a composited wind fields from dual-polarized images with rain corrections are validated against the measurements from SFMR.

The SAR-derived wind speeds from VH-polarized images and the SAR-derived wind directions from VV-polarized images are used as inputs in the GMF CMOD5.N. Then, the difference between SAR-measured and CMOD5.N-simulated NRCSs are computed. These samples are collocated with the measurements from SFMR in term of radar incidence angle. The relationship between the difference in NRCS and SFMR rain rate is shown in Figure 7, where the color bar represents the distance between the location of SFMR measurements and the TC's eye. Traditionally, the rainfall dominates at the inner circle of a TC (radius from TC eye less than 100 km) and the rainfall is relative weak at the outer circle of a TC (radius from TC eye greater than 100 km). The difference is divided into two regions: samples at the inner and outer circle. It is found that the difference in NRCS ranges from 0.5 dB to 5 dB (3 dB to 6 dB) when the distance from TC's eye is lower (larger) than 100 km. This is likely due to the fact that the wind speed is relatively smaller at the outer part of TCs, resulting in the rain cells highly affecting the sea surface roughness. According to these results, the identification of rain cells is implemented for the S-1

SAR dataset, see Figure 8, where they are marked as grey spots. Note that the sharp pattern of rain cells at the edge of S-1 swaths is due to the SAR winds derived from VH-polarized NRCS; see Figure 8d for the Hurricane Delta.

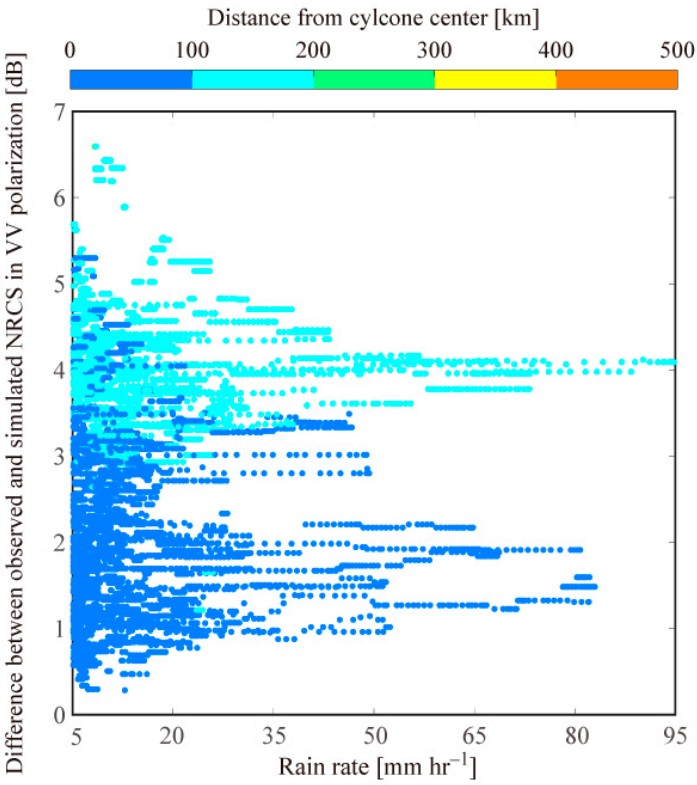

**Figure 7.** Relationship between the VV-polarized NRCS difference (SAR measurements versus CMOD5.N simulations) and SFMR rain rate. The color bar represents the distance between the location of SFMR measurements and the TC's center.

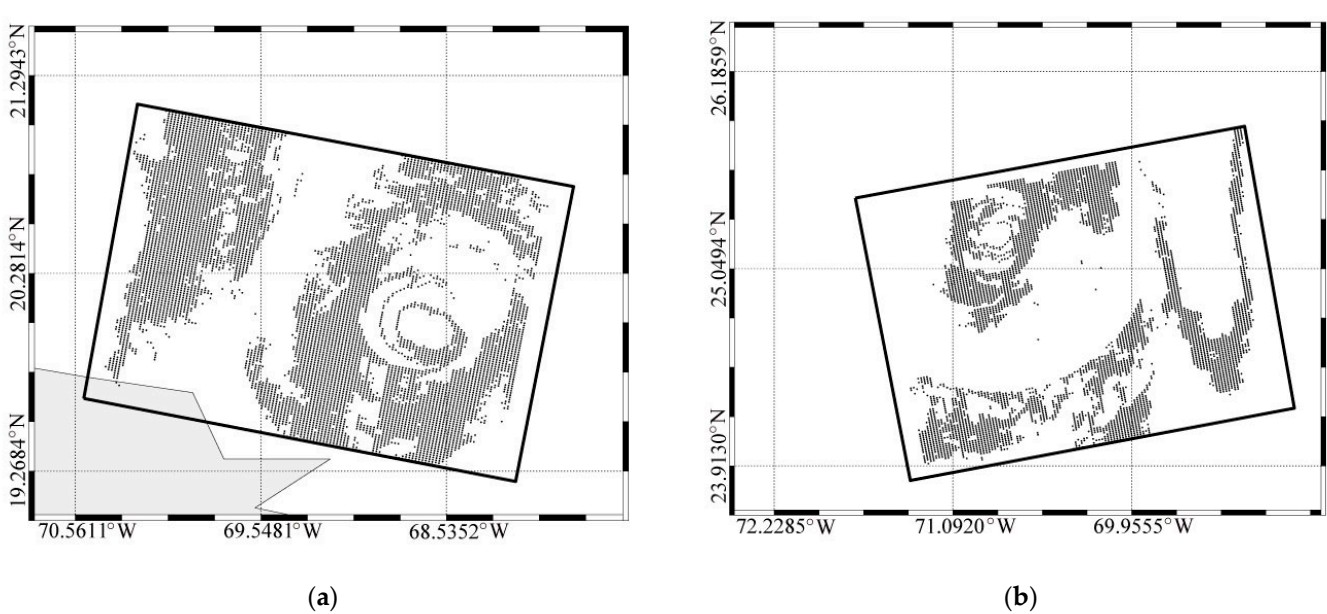

(**a**)                                    (**b**)

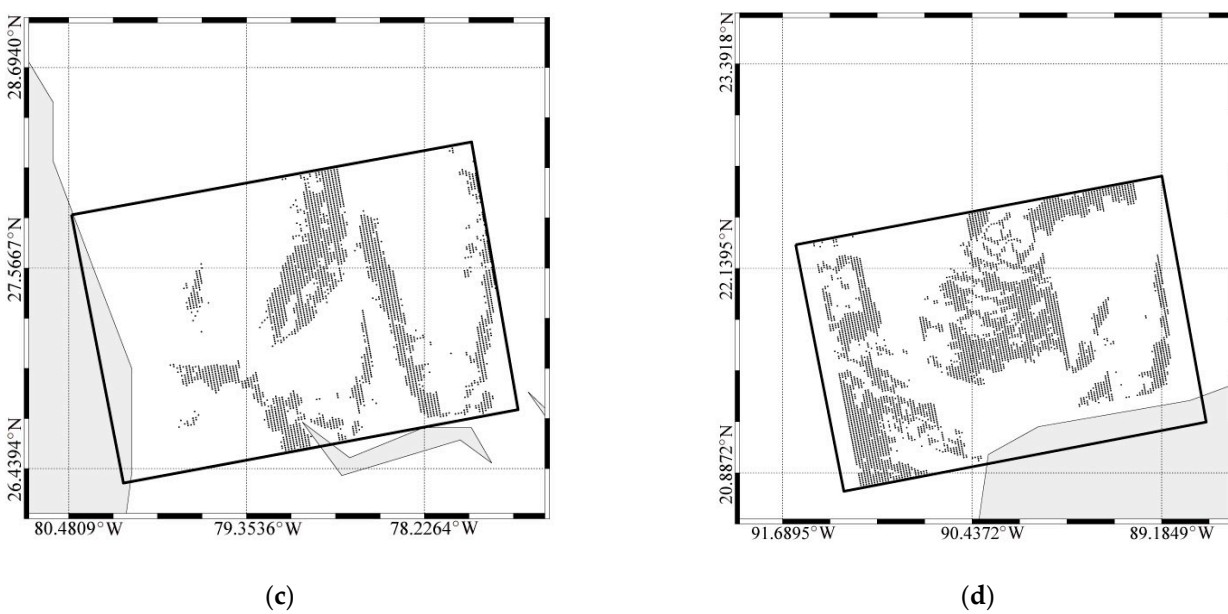

 (**c**) (**d**)

**Figure 8.** Maps of rain cell identification: (**a–d**) refer to the SAR scenes shown in Figure 1a–d.

The TC wind fields are composited by VH-polarized SAR-derived winds, VV-polarized SAR-derived winds and radial-vortex modeling winds. Each image is classified in a region distorted by the rain cells, where model-simulated winds are selected, and a rain-free region, where SAR-derived winds are selected. In particular, the VH-polarized SAR-derived winds at low-to-moderate condition (wind speed lower than 25 m s⁻¹) is replaced by the VV-polarized SAR-derived winds. Figure 9 shows wind speed in two radial profiles on the images of TC Irma overlaid by radial-vortex modeling winds at two flow directions: (a) 45° and (b) 135°, in which the red curve is the VH-polarized SAR-derived without rain, the blue curve is with rain and the black line is the model-simulated result. It is found that the pattern of the radial-vortex modeling wind profile performs well. Besides, the sharp gradient of VH-polarized SAR-derived wind in the radial profiles at the flow direction of 135° (see Figure 9b) is clearly observed, which is caused by the noise floor at the edge of swaths, e.g., 100 km radial distance away from TC eye. The composite winds speed maps obtained for the S-1 SAR dataset (see Figure 1) are shown in Figure 10. It can be noted that the pattern of extreme winds around the TC's eye (e.g., see Figure 10b relevant to the TC Dorian) is clearer than that in the VV-polarized and VH-polarized wind maps (e.g., see Figure 2b and Figure 3b, respectively). In addition, rain bands far away from the TC's eye (e.g., see Figure 10d relevant to the TC Delta) are also observed. The statistical analysis of composite winds validated against with SFMR measurements in term of flow directions of TC eyes is shown in Figure 11. In general, the accuracy of composited wind speed is improved after rain correction, e.g., the COR is greater than 0.7 at all flow directions. In particular, the RMSE of wind speed is reduced to be 1.8 m s⁻¹ with a 0.72 COR at the flow directions of 0°–30° and 330°–360°. This behavior is probably caused by the heavy rain in front of TC eyes. In this sense, we can conclude that the TC wind fields retrieved from dual-polarized SAR image are more reliable after rain correction.

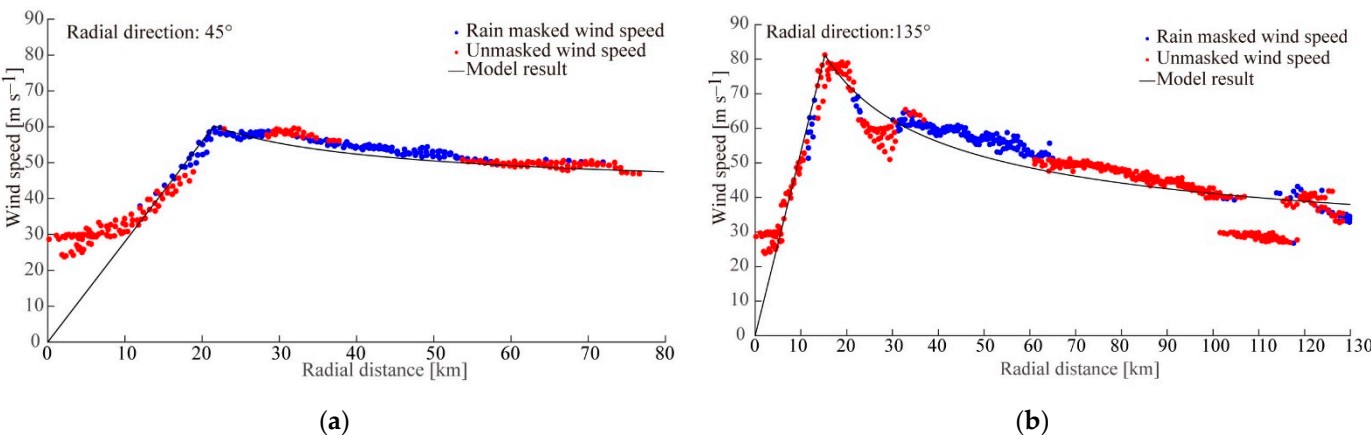

**Figure 9.** Wind speed in two radial profiles on the images of TC Irma overlaid by radial-vortex modeling winds at two flow directions: (**a**) 45° and (**b**) 135°. The red curve is the VH-polarized SAR-derived without rain, the blue curve is with rain and the black line is the model-simulated result.

**Figure 10.** Maps of composite SAR-derived winds: (**a**–**d**) refer to the SAR scenes shown in Figure 1a–d.

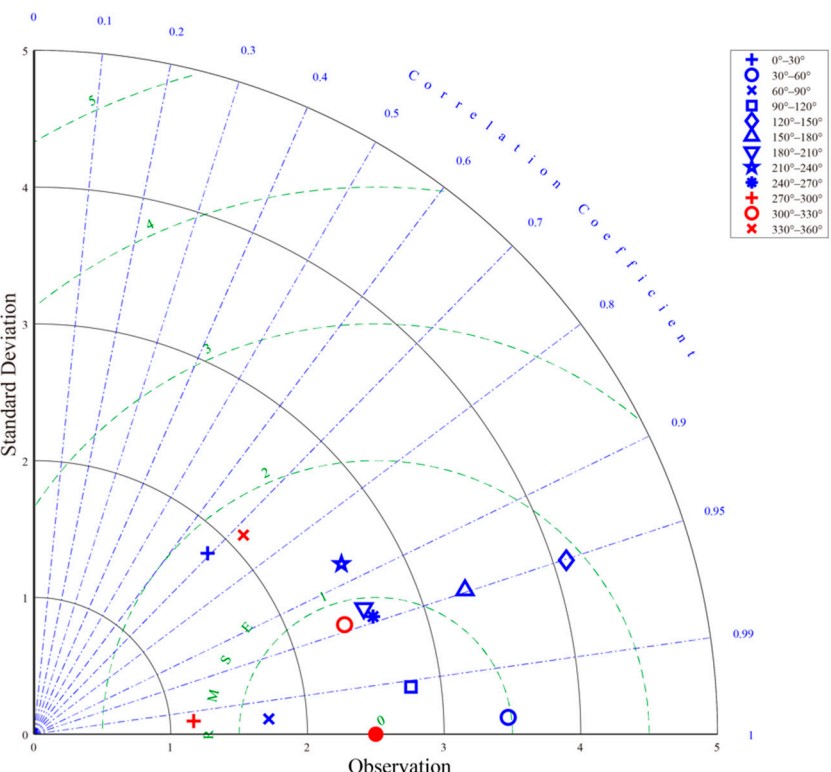

**Figure 11.** Taylor diagram of SAR-derived wind speeds versus SFMR observations.

## 4. Conclusions

Heavy rainfall, strong winds and extreme ocean waves are effects associated to TCs. Remote sensing tools as the microwave imager of the TRMM are promising for rainfall monitoring of the oceans. S-1 SAR mission operating at C-band has great capabilities providing the sea surface observations over large swath coverage and fine spatial resolution at both co- and cross-polarization. At present, sea winds associated to TCs could be inverted from VV- and VH-polarized SAR imagery by conventional GMFs that describe the semi-empirical relationships between NRCS and the wind field. Nonetheless, since rain cells directly modulate the ocean roughness resulting in changes of the measured NRCS, the wind retrieval accuracy reduces when heavy rainfall associated to TCs is in place.

In this study, a novel approach to wind retrieval from dual-polarized SAR imagery after rain correction is proposed. Four S-1 C-band SAR collected during TCs are considered, which are collocated with the wind speed and rain rate SFMR measurements. The SAR winds are retrieved from VH- and VV-polarized SAR images by S1IW.NR and CMOD5.N GMFs, respectively. The comparison between VH-polarized SAR-derived winds and the measurements of SFMR yields that maximum RMSE of wind speed is about 4 m s$^{-1}$ with a 0.45 COR ranged from 330° to 360°, whereas the RMSE of wind speed is less than 3 m s$^{-1}$ at other flow directions. The TC parameters are estimated by minimizing of the cost function, which relates the CMOD5.N-simulated NRCS taking VH-polarized SAR-derived wind speed < 20 m s$^{-1}$ with the SAR-measured NRCS. The retrieval TC intensity are compared with the best-track information of NOAA, indicating that the RMSE of central pressure and maximum wind speed is 15.6 hPa and 10.8 m s$^{-1}$, respectively. In the meantime, the difference between VV-polarized NRCS measured by the SAR and the NRCS simulated using the CMOD5.N GMF taking VH-polarized SAR winds as input is

related with the SFMR rain rate. Rain cells are identified on the basis that the NRCS difference ranges from 0.5 dB to 5 dB (3 dB to 6 dB) when the distance from TC's eye is lower (larger) than 100 km. Hence, a composite wind field is reconstructed by using dual-polarized SAR-derived winds and winds simulated by the parametric Holland model for rain-free and rain-affected regions, respectively. The validation results of composite winds against with SFMR show the RMSE < 2 m s$^{-1}$ and COR> 0.7 at all flow direction range, indicating that rain correction provides better performance in TC conditions.

As mentioned in [46], the automatic detection of rain cell necessitates complicated treatment especially for weak rain identification at regular sea state. Future work will include the improvement of rain identification and development of a new rain rate retrieval algorithm by means of a larger SAR dataset collected during TCs.

**Author Contributions:** Conceptualization, W.S. and Z.L.; methodology, W.S. and Z.L.; validation, W.S. and Z.L.; formal analysis, W.S., X.J. and J.Z.; investigation, W.S. and Z.L.; resources, W.S.; writing—original draft preparation, W.S., Z.L., F.N. and A.B.; writing—review and editing, W.S. and Z.L.; visualization, Z.L.; funding acquisition, W.S. All authors have read and agreed to the published version of the manuscript.

**Funding:** This research was funded by the National Natural Science Foundation of China, grant number 42076238, 42176012 and 42130402, and the Shanghai Frontiers Research Center of the Hadal Biosphere.

**Data Availability Statement:** Not applicable.

**Acknowledgments:** The S-1 SAR images were provided by ESA via https://scihub.copernicus.eu (accessed on 3 October 2022). The ERA-5 winds were accessed via http://www.ecmwf.int (accessed on 3 October 2022). We also appreciate the provision of NOAA for SFMR data and the information on hurricane best-tracks.

**Conflicts of Interest:** The authors declare no conflict of interest. The funders had no role in the design of the study; in the collection, analyses or interpretation of data; in the writing of the manuscript; or in the decision to publish the results.

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
