# Peer review of "Wind Field Retrieval with Rain Correction from Dual-Polarized Sentinel-1 SAR Imagery Collected during Tropical Cyclones"

_remotesensing, doi:10.3390/rs14195006_

Round 1

Reviewer 1 Report

  • Please find the comments in the attachment.

Author Response

We thank the reviewer for his constructive comments, which greatly helped us to improve this manuscript. We have provided point-by-point replies herein to carefully address these comments and suggestions.

Reply to Reviewer 1

General comment: Rain cells directly modulate the radar echoes, resulting the wind retrieval accuracy reduces when heavy rainfall is in place. To remove the impact of rain cells on wind retrieval, an approach for wind retrieval from dual-polarized SAR imagery after rain correction is proposed. The proposed approach sounds effective, and the results are promising. I recommend a minor revision of the paper.

Reply: We appreciate your positive suggestions. We have made careful and detailed modifications according to your suggestions in the revised manuscript.

Comment 1: The usage of English language and writing style must be improved. In large parts of the manuscript there is a clear lack of readability that makes difficult to get the messages. 

Reply: We have modified the manuscript structure for the reader to get the messages. Moreover, the usage of English language will be professional editing by the journal as the manuscript accepted.

Comment 2: Please modify the font of the equations.

Reply: We have modified the font of the formulas in the revised manuscript.

Comment 3: Line 37, Please add the full spell of COR.

Reply: We have added the full spell of COR in the revised manuscript.

Comment 4: Line 72, Please correct the sentence “As matter of fact,….”.

Reply: We have deleted the sentence in the revised manuscript.

Comment 5: Line 180, A recent study proposed by [17] shows the radar backscattering increases with rain rate at low-to-moderate winds. The basic principle is based on the complementary sensitivity of the SAR signal induced by rain cell, that is the sensitivity of NRCS to rain rate is more pronounced at higher incidence angles. Please correct or rewrite this sentence, I cannot find the connection between “the radar backscattering increases with rain rate” and “the sensitivity of NRCS to rain rate is more pronounced at higher incidence angles”

Reply: We rewrite the sentence in the revised manuscript.

Page 8, Line 212: A recent study proposed by [17] shows the radar backscattering increases with rain rate at low-to-moderate winds. The basic principle is based on the complementary sensitivity of the SAR signal induced by rain cell.

Comment 6: Line 193~197, Improve the description of the step-by-step method.

Reply: We Improve the description of the step-by-step method in the revised manuscript.

Page 8, Line 224: To summarize, the following three steps are needed: (1) wind speed retrieval from VH-polarized SAR image using GMF S1IW.NR at sub-scene of 3×3 km; (2) Prediction of the GMF CMOD5.N-simulated NRCS using SAR-derived wind direction and VH-polarized wind speed retrieved as in (1); (3) applying Equation (3) in term of radar incidence angle from SAR image to calculate the T; and (4) identifying the rain cells by T through whole image, such as the values smaller than T are assumed to be rain cells.

Comment 7: Line 196, How did you identify rain cells after “select the T”?

Reply: We had written identify rain cells part in the manuscript as follow:

Page 8, Line 224: To summarize, the following three steps are needed: (1) wind speed retrieval from VH-polarized SAR image using GMF S1IW.NR at sub-scene of 3×3 km; (2) Prediction of the GMF CMOD5.N-simulated NRCS using SAR-derived wind direction and VH-polarized wind speed retrieved as in (1); (3) applying Equation (3) in term of radar incidence angle from SAR image to calculate the T; and (4) identifying the rain cells by T through whole image, such as the values smaller than T are assumed to be rain cells.

Comment 8: Line 225, Please correct the expressions “”is the simulated NRCS using Holland winds.….”. NRCS cannot simulated by Holland wind directly.

Reply: We modified the sentence in the manuscript.

Page 9, Line 228: where, is the simulated NRCS by CMOD5.N GMF using Holland winds.

Comment 9: Fig. 6, Please add “Holland wind model” into the flowchart.

Reply: We have added “Holland wind model” into the flowchart.

Comment 10: Fig.1-Fig.3, Fig.8-Fig.10, the display range should be reduced in all these figures.

Reply: We modified the display range in the Fig.1-Fig.3, Fig.8-Fig.10 in the revised manuscript.

Reviewer 2 Report

The paper is dedicated to wind field retrieval with rain correction from dual-polarized Sentinel-1 SAR tropical cyclones imagery. Vertical-vertical (VV) and vertical-horizontal (VH) interferometric-wide swath imaging has been considered. A cost function is proposed in the article. It is based on the difference between the measured VV-polarized SAR normalized radar NRCS and the NRCS predicted using CMOD5.N forced with the wind speed retrieved by the VH-polarized SAR images using S1IW.NR GMF. The paper is written well. The results obtained are new, interesting, and valuable for the field. The results are clear, but their discussion should be provided in section Discussion, which should be organized in the paper. The paper also needs some corrections before its publication.

Corrections suggested.

1. The paper should have the following structure: Introduction, Materials and Methods, Results, Discussion, and Conclusions. Please, reorganize the paper in accordance with the required structure: https://www.mdpi.com/journal/remotesensing/instructions.

2. Please, add an empty line after Figure 5 caption.

3. Please, use ‘Figure’ instead of ‘Fig.’ in the text.

4. Please, increase resolution of Figure 9a and 9b.

5. Please, provide DOI for references in section References when available.

So, the paper needs minor revision.

Author Response

We thank the reviewer for their constructive comments, which greatly helped us to improve this manuscript. We have provided point-by-point replies herein to carefully address these comments and suggestions.

Reply to Reviewer 2

General comment: The paper is dedicated to wind field retrieval with rain correction from dual-polarized Sentinel-1 SAR tropical cyclones imagery. Vertical-vertical (VV) and vertical-horizontal (VH) interferometric-wide swath imaging has been considered. A cost function is proposed in the article. It is based on the difference between the measured VV-polarized SAR normalized radar NRCS and the NRCS predicted using CMOD5.N forced with the wind speed retrieved by the VH-polarized SAR images using S1IW.NR GMF. The paper is written well. The results obtained are new, interesting, and valuable for the field. The results are clear, but their discussion should be provided in section Discussion, which should be organized in the paper. The paper also needs some corrections before its publication.

Reply: We appreciate your positive suggestions. We have made careful and detailed modifications according to your suggestions in the revised manuscript.

Comment 1: The paper should have the following structure: Introduction, Materials and Methods, Results, Discussion, and Conclusions. Please, reorganize the paper in accordance with the required structure:https://www.mdpi.com/journal/remotesensing/instructions.

Reply: We have reorganized the revised manuscript follow the journal structure.

Comment 2: Please, add an empty line after Figure 5 caption.

Reply: We add an empty line after the caption of Figure 5 in the revised manuscript.

Comment 3: Please, use ‘Figure’ instead of ‘Fig.’ in the text.

Reply: We have replaced ‘Fig.’ with ‘Figure’ in the revised manuscript.

Comment 4: Please, increase resolution of Figure 9a and 9b.

Reply: The resolution of Figure 9a and 9b has been increased.

Comment 5: Please, provide DOI for references in section References when available.

Reply: We add the DOI for references in section References in the revised manuscript.

Reviewer 3 Report

In this paper, a new method was proposed to conduct wind retrieval from dual-polarized SAR imagery with a focus on rain correction. According to the comparison with the SFMR measurements, the accuracy of the composited wind speed here has been apparently improved after rain correction. Generally, this work by Shao et al. is meaningful for improving wind field retrieval under typhoon conditions. On the basis of my own evaluation, the manuscript can be considered for publication on Remote Sensing after a moderate revision. Please find my comments below.

(1) Most analyses of this paper were based on the data of four tropical cyclone cases. It is necessary to give reasons why the current four cases should be selected.

(2) Although the new method has improved the wind retrieval in these four cases, how about its performance in other tropical cyclones? How can the author prove the universality of their new method? If this is restricted by limited data that can be collected, the author should at least fully discuss this problem.

(3) The SFMR measurements were employed as validation data here. Descriptions about the quality and reliability of these data should not be absent.

(4) Most figures are of low quality and need to be redrawn. For example, Figures 1, 2, 3, 8, and 10 are made up of four independent subplots. And the subplots in each figure are in different sizes. Lots of blanks also exist in each figure. The font size in Figure 9 is too small to be recognized.

Specific comments:

L175: parametric -> parameter

L229: ... RMSE of central pressure and maximum wind speed is ...-> Should be RMSEs ... are ...

L240: ... is recently developed -> Should be ... was ...

L288: How is the direction defined? What directions do 0, 90°, 180°, 270° mean?

L295: It can be noted that that...-> Typo. Remove one that.

L342-344: This sentence is too long. Please rewrite it.

Author Response

We thank the reviewer for their constructive comments, which greatly helped us to improve this manuscript. We have provided point-by-point replies herein to carefully address these comments and suggestions.

Reply to Reviewer 3

General comment: In this paper, a new method was proposed to conduct wind retrieval from dual-polarized SAR imagery with a focus on rain correction. According to the comparison with the SFMR measurements, the accuracy of the composited wind speed here has been apparently improved after rain correction. Generally, this work by Shao et al. is meaningful for improving wind field retrieval under typhoon conditions. On the basis of my own evaluation, the manuscript can be considered for publication on Remote Sensing after a moderate revision. Please find my comments below.

Reply: We appreciate your positive suggestions. We have made careful and detailed modifications according to your suggestions in the revised manuscript.

Comment 1: Most analyses of this paper were based on the data of four tropical cyclone cases. It is necessary to give reasons why the current four cases should be selected.

Reply: Due to the clear cyclone eye in the SAR image and the sea surface wind and rain rate observations obtained by SFMR during the cyclone period, we chose the current four tropical cyclone cases.

Page3, Line 108: In this study, four S-1 SAR images acquired in dual-polarized (VV+VH) interferometric-wide (IW) swath imaging mode during TCs are considered with the apparent TC eyes, see Table. 1o=.

Comment 2: Although the new method has improved the wind retrieval in these four cases, how about its performance in other tropical cyclones? How can the author prove the universality of their new method? If this is restricted by limited data that can be collected, the author should at least fully discuss this problem.

Reply: Due to the lack of the SAR data, we only used four tropical cyclones to study the new method. In the future work, we will include the improvement of rain identification and development of a new rain rate retrieval algorithm by means of a larger SAR dataset collected during TCs.

Page 15, Line 531: Future work will include the improvement of rain identification and development of a new rain rate retrieval algorithm by means of a larger SAR dataset collected during TCs.

Comment 3: The SFMR measurements were employed as validation data here. Descriptions about the quality and reliability of these data should not be absent.

Reply: We add the new reference to show the quality and reliability of the SFMR measurements.

Page 3, Line 110: The SFMR provides high-quality and reliable observations including the wind speed and rain rate within and around these TCs [37].

Comment 4: Most figures are of low quality and need to be redrawn. For example, Figures 1, 2, 3, 8, and 10 are made up of four independent subplots. And the subplots in each figure are in different sizes. Lots of blanks also exist in each figure. The font size in Figure 9 is too small to be recognized.

Reply: We redraw the Figures 1, 2, 3, 8, and 10 and improve the font size in Figure 9 in the revised manuscript.

Comment 5: L175: parametric -> parameter

Reply: We modified it in the revised manuscript.

Comment 6: L229: “... RMSE of central pressure and maximum wind speed is ...”-> Should be “RMSEs ... are ...”

Reply: We modified it in the revised manuscript.

Comment 7: L240: “... is recently developed ”-> Should be “... was ...”

Reply: We modified it in the revised manuscript.

Comment 8: L288: How is the direction defined? What directions do 0, 90°, 180°, 270° mean?

Reply: The direction means the cyclone part. The 0 means the line from cyclone center to the north direction; 90° means the line from cyclone center to the east direction, 180°means the line from cyclone center to the south direction, and 270°means the line from cyclone center to the west direction.

Page 12 Line 459: Noted that the direction 0° means the line from cyclone center to the north direction; 90° means the line from cyclone center to the east direction;180°means the line from cyclone center to the south direction; and 270° means the line from cyclone center to the west direction.

Comment 9: L295: It can be noted that that...-> Typo. Remove one “that”.

Reply: We removed it in the revised manuscript.

Comment 10: L342-344: This sentence is too long. Please rewrite it.

Reply: We rewrote the sentences in the revised manuscript.

Page 15, Line 348: The validation results of composite winds against with SFMR show the RMSE < 2 m s-1 and COR> 0.7 at all flow direction range, indicating that rain correction provides better performance in TC conditions.